# Neuroprotective Actions of Cannabinoids in the Bovine Isolated Retina: Role of Hydrogen Sulfide

**DOI:** 10.3390/ph18010117

**Published:** 2025-01-17

**Authors:** Leah Bush, Anthonia Okolie, Jenaye Robinson, Fatima Muili, Catherine A. Opere, Sunny E. Ohia, Ya Fatou Njie Mbye

**Affiliations:** 1Department of Pharmaceutical Sciences, College of Pharmacy and Health Sciences, Texas Southern University, Houston, TX 77004, USA; lemwjb@gmail.com (L.B.); a.okolie9552@student.tsu.edu (A.O.); jenayerobinson@yahoo.com (J.R.); f.muili1867@student.tsu.edu (F.M.); sunny.ohia@tsu.edu (S.E.O.); 2Department of Pharmacy Sciences, School of Pharmacy and Health Professions, Creighton University, Omaha, NE 68178, USA; catherineopere@creighton.edu

**Keywords:** hydrogen sulfide, cannabinoids, retina, oxidative stress, lipid peroxidation

## Abstract

Both hydrogen sulfide and endocannabinoids can protect the neural retina from toxic insults under in vitro and in vivo conditions. **Purpose:** The aim of the present study was two-fold: (a) to examine the neuroprotective action of cannabinoids [methanandamide and 2-arachidonyl glycerol (2-AG)] against hydrogen peroxide (H_2_O_2_)-induced oxidative damage in the isolated bovine retina and (b) to evaluate the role of endogenously biosynthesized hydrogen sulfide (H_2_S) in the inhibitory actions of cannabinoids on the oxidative stress in the bovine retina. **Methods:** Isolated neural retinas from cows were exposed to oxidative damage using H_2_O_2_ (100 µM) for 10 min. When used, tissues were pretreated with methanandamide (1 nM–100 nM) and 2-AG (1–10 µM) for 30 min before a 10 min treatment with H_2_O_2_ (100 µM). In some experiments, retinas were pretreated with inhibitors of the biosynthesis of H_2_S [cystathionine β-synthase/cystathionine γ-lyase (CBS/CSE), aminooxyacetic acid, AOAA 30 µM, or 3-mercaptopyruvate sulfurtransferase (3MST), α-keto-butyric acid, KBA 1 mM] and the CB1-receptor antagonist, AM251 (100 nM) for 30 min before treatment with methanandamide (1 nM–100 µM). Enzyme immunoassay measurement of 8-epi PGF2α (8-isoprostane) levels was performed to assess lipid peroxidation in retinal tissues. **Results:** In the presence of H_2_O_2_ (100 µM), methanandamide (1 nM–100 µM) and 2-AG (1–10 µM) significantly (*p* < 0.001) blocked the H_2_O_2_-induced elevation in 8-isoprostane levels in the isolated bovine retina. In the presence of the CB1 antagonist AM251 (100 nM), the effect of methanandamide (1 nM) on the H_2_O_2_-induced 8-isoprostane production was significantly (*p* < 0.001) attenuated. While AOAA (30 µM) had no significant (*p* > 0.05) effect on the inhibition of H_2_O_2_-induced oxidative stress elicited by methanandamide, KBA (1 mM) reversed the neuroprotective action of methanandamide. **Conclusions:** The cannabinoids, methanandamide and 2-AG can prevent H_2_O_2_-induced oxidative stress in the isolated bovine retina. The neuroprotective actions of cannabinoids are partially dependent upon the activation of the CB1 receptors and endogenous production of H_2_S via the 3-MST/CAT pathway.

## 1. Introduction

Cannabinoids are amides and esters of long-chain polyunsaturated fatty acids that are found in the marijuana plant and in a variety of mammalian tissues [1,2]. These lipid mediators, their receptors, and the enzymes that degrade them are involved in various biological activities throughout the CNS in areas such as memory and cognition, neuronal function and survival, immune response, anxiety and depression, and addiction [2,3,4]. CB1 and CB2 receptors are found in most retinal neurons and astrocytes including amacrine cells, photoreceptors, horizontal cells, Müller glia, bipolar cells, and retinal ganglion cells [5]. The presence of CB1 receptors in the neural retina indicates that cannabinoids play a regulatory role in retinal circuitry by affecting the release of other neurotransmitters. In the eye, the endocannabinoid system has been shown to play a role in regulating aqueous humor dynamics and IOP and in protecting the retina against neuronal damage [3,6,7,8,9]. There is evidence of an alteration in endocannabinoid levels in the retina of glaucoma, age-related macular degeneration (ARMD), and diabetic retinopathy (DR) patients [10,11], validating a role of the endocannabinoid system in pathophysiological processes associated with neuronal degeneration in the eye. In the CNS, endogenous cannabinoids and H_2_S have been shown to elicit similar physiological and pharmacological actions including their neuroprotective effects in neurons against injury. These lipid-soluble mediators are produced via calcium-dependent processes leading to their on-demand release [5,12,13,14]. Cannabinoids and H_2_S have also been reported to inhibit glutamate release in the retina, and both agents can regulate voltage-gated ion channels in neuronal tissues [5,15,16,17,18,19]. Additionally, endogenous cannabinoids and intramurally produced H_2_S have been shown to act as antioxidants [20,21] and as anti-inflammatory agents in oxidative and inflammatory injury in the retina [22,23]. The most abundant endocannabinoid, 2-arachidonoyl glycerol (2-AG), is present in ocular tissues with its concentration varying in normal and glaucomatous human eyes and in patients with diabetic retinopathy and age-related macular degeneration [10]. The synthetic cannabinoid, methanandamide (methAEA), as a selective agonist acts on CB1 receptors present in ocular tissues such as the cornea, iris–ciliary body, and the retina [24]. Interestingly, some of the pharmacological actions of H_2_S have been attributed to its ability to activate transient receptor potential vanilloid (TRPV) channels, a family of highly calcium-selective ion channels that can also be activated by the endogenous cannabinoid, anandamide (AEA) [25,26]. In ocular tissues, H_2_S and cannabinoids have also been found to relax ocular blood vessels, an effect that can enhance overall blood supply to the anterior and posterior segments of the eye [27,28,29,30,31].

Since both cannabinoids and H_2_S have been reported to possess neuroprotective actions in several tissues/organs, the aims of the present study are two-fold: (a) to assess the neuroprotective activity of cannabinoids (2-AG and methAEA) under conditions of H_2_O_2_-induced oxidative damage in the isolated bovine retina and (b) to evaluate the role of intramurally synthesized H_2_S in the inhibitory effects of cannabinoids on the oxidative stress response in this tissue. Parts of this paper have been communicated in an abstract form [32].

## 2. Results

In a previous study, we reported that H_2_O_2_ (30–300 µM) produced a concentration- and time-dependent increase in the production of 8-isoprostane in isolated bovine neural retina [33]. Since H_2_O_2_ (100 µM) produced a 30% increase in 8-isoprostane concentration over basal levels [33], we selected the same oxidant concentration to induce oxidative stress in the present study.

### 2.1. Effects of Cannabinoids on Lipid Peroxidation in the Bovine Retina

Both endogenous cannabinoid compounds and synthetic agonists of cannabinoid receptors have been implicated in defense of neurons against oxidative stress [34,35,36]. In a series of experiments, we investigated the effect of cannabinoid pretreatment on H_2_O_2_-induced oxidative stress in isolated bovine retinal tissues. Methanandamide and the endogenous cannabinoid, 2-AG were the compounds utilized in the present study [5,27]. Retinal tissues were incubated in different concentrations of methanandamide (1 nM–100 µM) before treatment with H_2_O_2_ (100 µM). After 30 min of treatment, methanandamide (1 nM) completely reversed the H_2_O_2_-induced increase in 8-isoprostane levels (Figure 1).

In contrast, higher concentrations of methanandamide (10 µM and 100 µM) did not affect the H_2_O_2_-induced increase in 8-isoprostane levels (Figure 1). Indeed, in the presence of methanandamide (100 µM), there was a 38% (*p* < 0.01) increase in H_2_O_2_-induced-8-isoprostane levels when compared to the control (tissues treated with H_2_O_2_ alone). We next examined the effect of 2-AG on H_2_O_2_-induced lipid peroxidation in the isolated retina. At an incubation time of 30 min, 2-AG (3 µM) significantly (*p* < 0.001) attenuated H_2_O_2_-induced 8-isoprostane production, whereas 10 µM of this endocannabinoid caused a significant (*p* < 0.001) 32% increase in H_2_O_2_-induced 8-isoprostane levels over the control (tissues treated with H_2_O_2_ alone) (Figure 2).

### 2.2. Involvement of CB1 Receptors and Intramurally Generated H_2_S in Cannabinoid-Mediated Neuroprotection in the Bovine Retina

There is evidence that the protective pharmacological actions of cannabinoids in the retina could be facilitated by the activation of cannabinoid receptors, particularly CB1, and by an effect on TRPV channels [3,6]. In a series of experiments, we investigated the effect of a CB1-receptor antagonist, AM251, on the methanandamide-induced attenuation of H_2_O_2_-induced 8-isoprostane production. On its own, AM251 had no significant effect on retinal isoprostane production (Figure 3). As shown in Figure 3, treatment of tissues with AM251 (100 nM) significantly (*p* < 0.001) blocked the inhibitory effect of methanandamide on H_2_O_2_-induced 8-isoprostane production.

Although the endocannabinoid system has been reported to interact with NO, CO, and H_2_S in the heart, renal tissues, GI tract, vasculature, adipose tissue, and spinal cord [37,38,39,40], its potential interaction with pathways leading to H_2_S production in the retina is unknown. To examine the role of H_2_S on the pharmacological action of methanandamide, isolated retinae were treated with inhibitors of H_2_S biosynthesis before exposure to methanandamide and oxidative damage [H_2_O_2_ (100 µM)]. As shown in Figure 4, an inhibitor of the biosynthetic enzymes for H_2_S CBS/CSE (AOAA 30 µM) had no effect on basal isoprostane production or on the methanandamide (1 nM)-induced attenuation of H_2_O_2_-induced oxidative damage.

Interestingly, an inhibitor of the 3-MST pathway for the biosynthesis of H_2_S, KBA (1 mM), which had no effect on basal 8-isoprostane production, completely reversed responses elicited by methanandamide (1 nM) (Figure 4).

## 3. Discussion

It is well-known that cannabinoids can serve as lipid neuromodulators in various mammalian tissues and organs [2,41,42,43,44]. In fact, there is evidence that the endogenous cannabinoid system can possess therapeutic potential in areas including memory and cognition, neuronal function and survival, immune response, anxiety and depression, and addiction [2,3,45]. In the cardiovascular system, the endocannabinoid system has been reported to play an important role in the maintenance of vascular smooth muscle tone, whereas in the CNS, it has been demonstrated to act as a neurotransmitter and neuromodulator at synapses [1,42]. The cannabinoid system has been reported to interact with well-established gaseous transmitters such as NO and CO [39,40,46]. There is evidence that activation of CB1 receptors elicited an inhibitory effect on the biosynthesis of NO in astrocytes, endothelial cells, microglia, and neurons [40]. Treatment with CO-releasing compounds has been found to enhance the analgesic effect of CB2-receptor agonists and the expression of CB2 receptors in mouse models of inflammatory pain and diabetic neuropathy [39,47]. Administration of a CB1-receptor agonist over two weeks augmented H_2_S production in isolated perivascular adipose tissue and reduced mitochondrial H_2_S oxidation [37]. Although there are emerging studies focused on interpreting the nature of crosstalk between the endocannabinoid system and gaseous mediators, there is a paucity of knowledge concerning the ability of these two systems to interact with each other in ocular tissues.

The cytoprotective actions of the endocannabinoid system have been extensively studied in numerous systems [2,41,42,43,44]. Furthermore, cannabinoids have also been reported to exert a neuroprotective action on neurons [48]. For instance, CB2-receptor agonists and inhibitors of the metabolism of AEA and 2-AG have been reported to prevent neurodegenerative processes in a mouse model of traumatic brain injury [49]. In the cardiovascular system, cannabinoids have been shown to protect the heart by decreasing myocardial infarct size in a rat model of ischemia [50]. Evidence in the literature also supports a neuroprotective role for the endogenous cannabinoid system in ocular tissues [3,5,9,43]. Indeed, elevated retinal anandamide levels have been associated with ARMD [11]. In a cellular model of ARMD, blockade of CB1 receptors protected retinal pigment epithelium cells from oxidative damage [21]. Anandamide and 2-AG have also been reported to play a role in the modulation of the innate immune response in human retinal Müller glia cells’ defense against inflammation during human immunodeficiency virus (HIV) infection [22]. It appears that the HIV infection induces retinal neurodegeneration by increasing inflammation, resulting in retinal dysfunction. In a 2014 study, Krishnan and colleagues found that both anandamide and 2-AG reduced retinal inflammation and prevented cell death [22,43]. In the present study, we observed neuroprotection elicited by the endogenous cannabinoid, 2-AG, and the nonhydrolyzable synthetic anandamide analogue, methanandamide, against H_2_O_2_-induced lipid peroxidation in the isolated neural retina. Pretreatment of isolated bovine retinal tissues with a low concentration of methanandamide (1 nM) prevented the H_2_O_2_-induced increase in 8-isoprostane levels. In contrast, pretreatment with higher concentrations of methanandamide (10 nM–100 µM) for 30 min elicited a significant increase in the production of 8-isoprostane in the bovine retina. It is pertinent to note that the observed increase in oxidative stress induced by higher concentrations of methanandamide was unexpected. Our observation is supported by reports from other laboratories where AEA was found to induce cell death of primary neurons in vitro [51,52]. It was previously reported that polyunsaturated fatty acids can increase intracellular oxidative stress, including lipid peroxidation [53,54]. Indeed, AEA has been found to enhance the production of 8-isoprotane in head and neck squamous cell carcinoma cells and microglial cells treated with lipopolysaccharide (LPS) [55,56]. Taken together, these observations indicate that methanandamide is capable of acting as a prooxidant when administered in high concentrations in the isolated bovine retina.

In a series of experiments, exposure of retinal tissues to low concentrations of 2-AG (1–3 µM) for 30 min significantly blocked the H_2_O_2_-dependent increase in 8-isoprostane levels, whereas higher concentrations of the endocannabinoid enhanced 8-isoprostane production in the neural retina. The observed enhancement of H_2_O_2_-induced 8-isoprostane production is similar to responses elicited by methanadamine indicating that 2-AG may also be capable of acting as a prooxidant [55,56]. In summary, the observed pharmacological actions of methanandamide and 2-AG in the bovine retina support the dualistic nature of cannabinoids in their ability to regulate oxidative stress.

The pharmacological actions of cannabinoids have been demonstrated to involve the activation of cannabinoid receptors and TRPV channels [3,5,43]. For example, intravitreal injection of methanandamide was found to rescue RGCs from retinal ischemia-reperfusion injury via activation of CB1 receptors and TRPV1 channels in an in vivo IOP-reperfusion model of glaucoma [6]. The IOP-reducing effects of AEA and other cannabinoids are partially mediated by the activation of CB1 receptors in ocular tissues [3,5,43,57]. In the present study, the CB1-receptor antagonist, AM251, partially blocked the inhibitory effect of methanandamide (1 nM) on 8-isoprotane production in the bovine retina. Our finding that blockade of CB1 receptors reversed the neuroprotection caused by methanandamide supports the observation made by other investigators [5,57] that these receptors play a role in the pharmacological actions of methanandamide in the retina.

The interaction between cannabinoids and gaseous neurotransmitters such as NO, CO, and H_2_S has been reported in several tissues and organs [37,39,40]. For instance, cannabinoids can alter the synthesis and activity of NO in neurons, astrocytes, and cardiac cells under normal and damaging conditions [40]. The anti-nociceptive activity of CO and its production can also be modulated by CB2-receptor agonists in animal models of neuropathic pain [39,46]. The production of H_2_S has been reported to be increased by a cannabis sativa extract, MFF, in the colon, and by arachidonyl-2′-chloroethylamide (ACEA) in the perivascular adipose tissue of rats [37,38]. However, to the best of our knowledge, there are no reports of the potential interaction of the cannabinoid system and gaseous mediators such as H_2_S in ocular tissues. In the present study, we observed that inhibition of the 3MST/CAT pathway for the biosynthesis of H_2_S, completely reversed the methanandamide-induced neuroprotection, indicating a role for the gas produced by the mitochondrial pathway in the antioxidant actions of this cannabinoid in the retina. Indeed, there is evidence that activation of CB1 receptors in perivascular adipose tissue decreased mitochondrial H_2_S oxidation [37]. Furthermore, the neuroprotective actions of L-cysteine in the isolated bovine retina were reversed in the presence of an inhibitor of mitochondrial H_2_S biosynthesis, KBA [33]. It appears that the pathway for the biosynthesis of H_2_S during the neuroprotective actions of H_2_S-releasing compounds and the cannabinoids are similar since they both involve the mitochondrial 3MST/CAT pathway. Taken together, these observations support the role of the mitochondrial-derived gas in the neuroprotective action of cannabinoids in the isolated bovine retina. It is pertinent to note that inhibition of CBS and CSE by aminooxyacetic acid did not affect the neuroprotective action of methanandamide on H_2_O_2_-induced oxidative damage. Based on the present finding, it appears that only the 3MST/CAT pathway of H_2_S biosynthesis may be involved in the neuroprotective action of cannabinoids in the isolated bovine retina.

A major pharmacological action of cannabinoids in the eye is an effect on aqueous humor dynamics leading to a reduction in intraocular pressure [58]. Cannabinoids have been reported to produce ocular hypotensive actions in both experimental animals and humans and to exhibit potential neuroprotective effects in the retina [58]. The observed neuroprotective action of cannabinoids in the present study and the potential involvement of H_2_S in its response supports the evidence reported by other investigators that these compounds may have therapeutic utility in the treatment of some eye diseases.

## 4. Methods

### 4.1. Chemicals

H_2_O_2_, aminoxyacetic acid (AOAA), and α-ketobutyric acid (KBA) were purchased from Sigma Chemical (St. Louis, MO, USA). Methanandamide, 2-AG, AM251, and an 8-Isoprostane enzyme-linked immunoassay kit was purchased from Cayman Chemical (Ann Harbor, MI, USA). All chemicals or drug test agents were freshly prepared immediately before being used for experiments. A stock solution of methanandamide was prepared in 70% ethanol. Stock solutions of SB366791 and AM251 were prepared in 100% DMSO, and stock solutions of all other compounds were prepared in deionized water.

### 4.2. Tissue Preparations

Studies were performed using cow eyeballs obtained from a local slaughterhouse (Fisher Ham and Meat Company, Houston, TX, USA) and transported to the laboratory on ice. As described previously [33], a cut was made along the limbus of each eye, and the vitreous humor and lens were delicately removed. The neural retina was then isolated by gentle removal from its attachment to the pigment epithelium in the posterior segment of the eye. The retina was immediately immersed in warm oxygenated (95% O_2_; 5% CO_2_) Krebs buffer solution containing the following (millimolar): potassium chloride, 4.8; sodium chloride, 118; calcium chloride, 1.3; potassium dihydrogen phosphate, 1.2; sodium bicarbonate, 25; magnesium sulfate, 2.0; and dextrose, 10 (pH 7.4).

### 4.3. 8-Isoprostane ELISA Assay

The methodology for extraction of 8-isoprostane from retinal tissues was performed in the same manner as described by our laboratory and other investigators [59,60,61,62] with some modifications. Briefly, isolated bovine retinae were equilibrated in oxygenated Krebs solution at 37 °C for 20 min and then transferred and incubated in another beaker containing Krebs solution in the presence and absence of methanandamide (1 nM–100 µM) or 2-AG (1–10 µM) for 30 min. To determine the pharmacological actions of cannabinoids against H_2_O_2_-induced 8-isoprostane production, retinal tissues were subjected to 10 min exposure to H_2_O_2_ (100 µM) following a 30 min incubation with test compounds. The concentration of H_2_O_2_ selected for the oxidative insult was according to a report by Bush et al. [33]. For mechanistic studies, tissues were pretreated with the CB_1_-receptor antagonist AM251 (100 nM) for 30 min before treatment with methanandamide (1 nM) to determine the role of these known cannabinoid targets in the methanandamide-mediated response. Retinal tissues were also treated with the CBS/CSE inhibitor, aminooxyacetic acid (AOAA, 30 µM), and the 3-MST inhibitor, ketobutyric acid (KBA, 1 mM) for 30 min before treatment with methanadamide (1 nM) in studies aimed to investigate the involvement of the H_2_S pathway in cannabinoid-mediated responses. After incubation, tissues were homogenized in 0.1 M phosphate buffer, pH 7.4 containing 1 mM EDTA and 0.005% BHT (1 mL/100 mg tissue), and centrifuged at 3000 r.p.m. for 10 min at 5 °C. The supernatant was collected and purified using potassium hydroxide. The 8-isoprotane in the supernatant was then extracted from purified samples using solid phase extraction cartridges and an ethyl acetate/methanol (99:1) mixture [6]. The 8-isoprotane in the sample was further concentrated by evaporating the ethyl acetate/methanol solution under N_2_ gas. The EIA buffer was used to re-suspend concentrated 8-isoprotane before running the ELISA assay. The protein content was determined from the unpurified supernatant using a Cayman protein determination kit (Cayman Chemical, Ann Arbor, MI, USA).

### 4.4. Data Analysis

Results obtained from the various experiments were expressed as 8-isoprotane concentrations per milligram soluble protein (pg/mg protein). All values are reported as means ± S.E.M. Significance of differences between values obtained in the control and drug-treated preparations were evaluated using one-way ANOVA. Time- and concentration-dependent effects were determined using two-way ANOVA. Differences with *p* values < 0.05 were accepted as statistically significant.

## 5. Conclusions

We conclude that cannabinoids such as methanandamide and 2-AG can protect the isolated bovine retina from H_2_O_2_-induced lipid peroxidation. The neuroprotection provided by methanandamide was dependent upon the activation of CB1 receptors and the endogenous production of H_2_S from the 3MST/CAT pathway. The observation that a pathway leading to the biosynthesis of H_2_S is involved in the neuroprotective action of cannabinoids is novel and merits further investigation.

## Figures and Tables

**Figure 1 pharmaceuticals-18-00117-f001:**
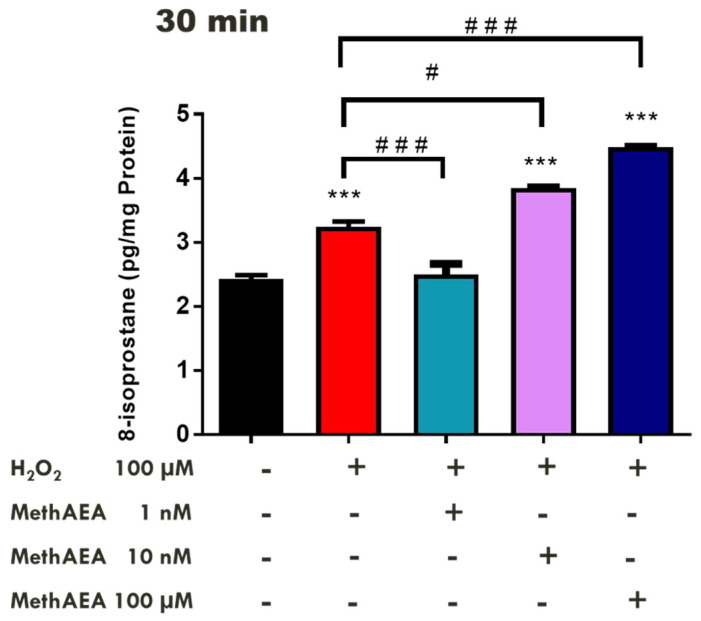
Concentration-dependent effect of methanandamide on H_2_O_2_-induced 8-isoprostane production in isolated bovine retina. Each value represents the mean ± SEM for *n* = 12; *** *p* < 0.001 significantly different from the control; ^#^ *p* < 0.05, ^###^ *p* < 0.001 significantly different from H_2_O_2_-treated tissues.

**Figure 2 pharmaceuticals-18-00117-f002:**
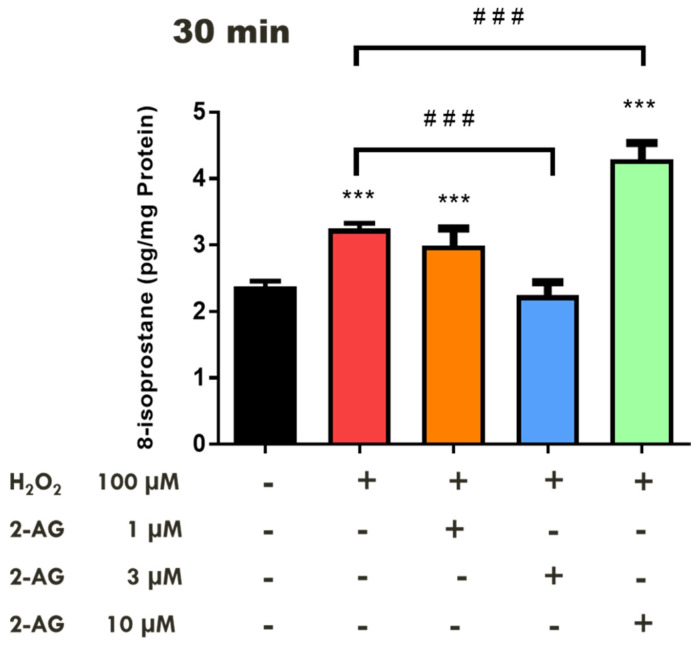
Concentration-dependent effect of 2-arachidonyl glycerol on H_2_O_2_-induced 8-isoprostane production in isolated bovine retina. Each value represents the mean ± SEM for *n* = 12. *** *p* < 0.001 significantly different from the control; ^###^ *p* < 0.001 significantly different from H_2_O_2_-treated tissues.

**Figure 3 pharmaceuticals-18-00117-f003:**
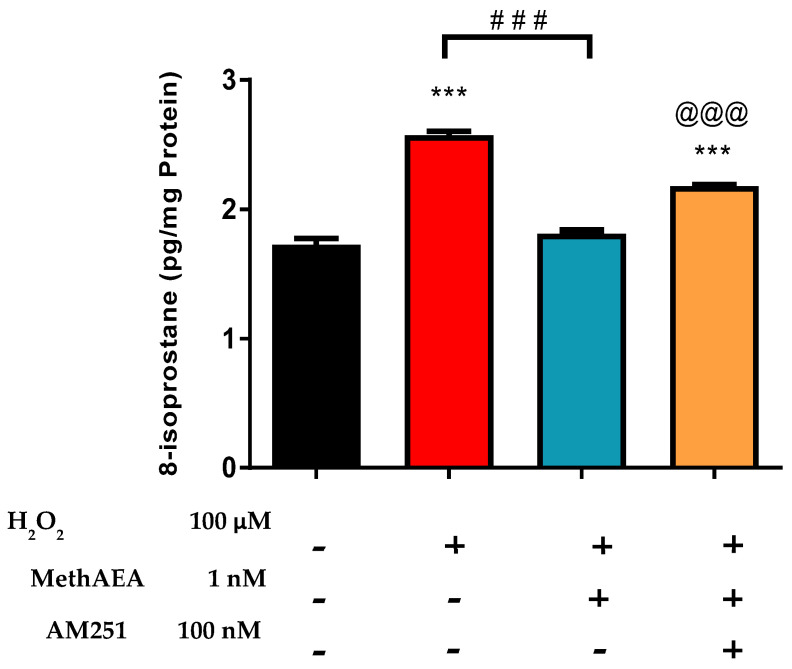
Role of the CB1 receptor and TRPV channels in methanandamide-mediated effects on H_2_O_2_-induced 8-isoprostane production in isolated bovine retina. Vertical bars represent mean ± S.E.M. *n* = 12; *** *p* < 0.001 significantly different from the control; ^###^ *p* < 0.001 significantly different from H_2_O_2_-treated tissues; ^@@@^ *p* < 0.001 significantly different from methanandamide-treated tissues.

**Figure 4 pharmaceuticals-18-00117-f004:**
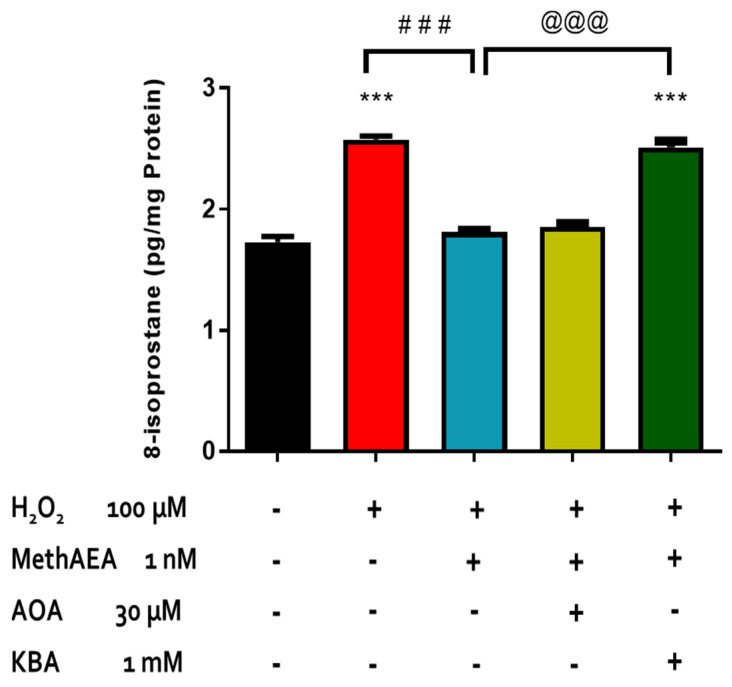
Role of CBS/CSE and the 3MST pathway in methanandmide-mediated neuroprotection in isolated bovine retina. Vertical bars represent mean ± S.E.M. *n* = 12. *** *p* < 0.001 significantly different from the control; ^###^ *p* < 0.001 significantly different from H_2_O_2_-treated tissues; ^@@@^ *p* < 0.001 significantly different from methanandamide-treated tissues.

## Data Availability

Data is contained within the article.

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
