# Peer review of "Neuroprotective Actions of Cannabinoids in the Bovine Isolated Retina: Role of Hydrogen Sulfide"

_pharmaceuticals, 2025, doi:10.3390/ph18010117_

Round 1
Reviewer 1 Report
Comments and Suggestions for Authors
The MS entitled “Neuroprotective actions of cannabinoids in the bovine isolated retina: role of hydrogen sulfide” was reviewed. The aim of the present study was to investigate the neuroprotective action of two cannabinoids against hydrogen peroxide (H2O2)-induced oxidative stress in retina, and (b) to evaluate the role of intramurally-synthesized hydrogen sulfide (H2S) in in the inhibitory actions of cannabinoids on the oxidative stress in this tissue. The MS is well composed and properly written while a concise scientific pattern was followed. The Results seems to fine and minor corrections are needed. My comments on the MS are as follow.
1. 25, 26; the phrase (b) to evaluate the role of intramurally-synthesized hydrogen sulfide (H2S) in in the inhibitory actions of cannabinoids on the oxidative stress in this tissue should be rephrase for better understanding.
2. Line 26; “oxidative stress in this tissue” which type of tissue the authors pointed to, it should be mentioned here.
3. Line 29; the author used “10-minute insult with H2O2” it would be better if replace the word “insult” by treatment.
4. Line 33; reduce space before word Enzyme.
5. Line 64; the font size of word “H2S” looks different from other text, make it uniform.
6. Line 81; reduced space before word “parts”.
7. Line 95 needs corrections. Prior to H2O2 or followed by H2O2induction?
8. Figure 1 should be re-adjusted by reducing the pixel resolution.
9. Line 103; reduce space before the word “indeed”.
10. Figure 2; resolution should be decrease.
11. Line 120; reduce space after reference.
12. Similarly, pixel ize of figure 3 and 4 should be reduced to normal.
13. In method section the protocols of “Tissue preparation” is presented without reference.
14. How 3MST/CAT 229 pathway for the biosynthesis of H2S is inhibited?
15. Line 283, how 8-isoprotane was purified and quantified? This need to be in detail.
16. H2S needs correction (line 163-164). Also, H2O2 in line 264. Check all the MS for such mistakes. Specially the discussion section.
17. Figures Keys do not fit in the whole figure scheme. Kindly align and adjust it with figure presentation.
18. It was strange that 1nM concentration was active inhibitor f oxidative stress while higher concentrations were not. This need to be discussed.
19. What the author expects from this study in the context of future endurance, it should be mentioned in conclusion section.
Comments on the Quality of English LanguageNo issue found.
Author Response
Comment 1: Figure 3, labels are not clear to read
MethAEA concentration is missing
Response 1: We have corrected the errors identified by the Reviewer in the Figure 3 labels and Legend.
Comment 2: In the introduction where general benefits/issues on cannabionoids are discussed, focus on methanandamide and 2-AG can be included in a few sentences/small section. Also, the future potential therapeutic implications of the findings can be included in the conclusion.
Response 2: As suggested by the Reviewer, we have included a few sentences on 2-AG and methanadamide in the Introduction section (lines 72-78) and have named them as cannabinoids to be tested in the present study (line 86). We have also discussed the potential therapeutic implications of our findings in the Discussion section (lines 256-263).
Comment 3: H2O2-mediated effects are observed in cancer and inflammation, a discussion on the future prospects of the findings on diverse therapeutic areas can be useful.
Response 3: Since we used oxidative stress as a tool to study neurotoxic damage and its prevention by cannabinoids in the present study, we are of the opinion that discussion the biological actions of H2O2 on diseases such as cancer and inflammation will be outside the scope of our manuscript.

Reviewer 2 Report
Comments and Suggestions for Authors
The article by Nush et al. entitled ‘Neuroprotective actions of cannabinoids in the bovine isolated retina: role of Hydrogen Sulfide’ highlights that cannabinoids, methanandamide and 2-AG can prevent Hydrogen peroxide-induced oxidative stress in the bovine isolated retina. The activation of CB1 receptors and H2S production contributes to the protective roles of cannabinoids tested.
The article is well written, the study is well designed and data confirms the hypothesis. However, minor spelling errors (e.g. line 25) and figure legends can be corrected to further improve the article. The minor suggestsions are listed below,
11. Figure 3, labels are not clear to read
MethAEA concentration is missing
22. In the introduction where general benefits/issues on cannabionoids are discussed, focus on methanandamide and 2-AG can be included in a few sentences/small section. Also, the future potential therapeutic implications of the findings can be included in the conclusion.
33. H2O2-mediated effects are observed in cancer and inflammation, a discussion on the future prospects of the findings on diverse therapeutic areas can be useful.
Comments on the Quality of English Language
Well written with minor errors
Author Response
Comment 1: 25, 26; the phrase (b) to evaluate the role of intramurally-synthesized hydrogen sulfide (H2S) in in the inhibitory actions of cannabinoids on the oxidative stress in this tissue should be rephrase for better understanding.
Response 1: We have rephrased the sentence to reflect the fact that the role of endogenously biosynthesized H2S will be studied (line 25).
Comment 2: Line 26; “oxidative stress in this tissue” which type of tissue the authors pointed to, it should be mentioned here.
Response 2: We have replaced “in this tissue” with bovine retina (line 27).
Comment 3: Line 29; the author used “10-minute insult with H2O2” it would be better if replace the word “insult” by treatment.
Response 3: We have replaced “insult” with the word, treatment (line 29).
Comment 4: Line 33; reduce space before word Enzyme.
Response 4: We have deleted the space before the word, Enzyme (line 34).
Comment 5: Line 64; the font size of word “H2S” looks different from other text, make it uniform.
Response 5: We have changed the font size of H2S as requested (line 66).
Comment 6: Line 81; reduced space before word “parts”.
Response 6: We have removed the space before the word, Parts (line 90).
Comment 7: Line 95 needs corrections. Prior to H2O2or followed by H2O2induction?
Response 7: We have changed the sentence to reflect the addition of H2O2 before the treatment with methanadamide (line 105).
Comment 8: Figure 1 should be re-adjusted by reducing the pixel resolution.
Response 8: We have reduced the pixel font size in Figure 1
Comment 9: Line 103; reduce space before the word “indeed”.
Response 9: We have deleted the space before, Indeed (line 113)
Comment 10: Figure 2; resolution should be decrease.
Response 10: We have reduced the pixel font size for Figure 2
Comment 11: Line 120; reduce space after reference.
Response 11: We have deleted space after reference citation (line 130).
Comment 12: Similarly, pixel ize of figure 3 and 4 should be reduced to normal.
Response 12: We have reduced the pixel font size for Figures 3 and 4.
Comment 13: In method section the protocols of “Tissue preparation” is presented without reference.
Response 13: We have included a citation for the the tissue preparation section (line 277).
Comment 14: How 3MST/CAT 229 pathway for the biosynthesis of H2S is inhibited?
Response 14: Keto-butyric acid was used to inhibit the 3MST/CAT pathway.
Comment 15: Line 283, how 8-isoprotane was purified and quantified? This need to be in detail.
Response 15: The information requested by the Reviewer was described by in the Methods section (lines 305 – 309).
Comment 16: H2S needs correction (line 163-164). Also, H2O2in line 264. Check all the MS for such mistakes. Specially the discussion section.
Response 16: We have corrected the abbreviations for H2S and H2O2 in the manuscript.
Comment 17: Figures Keys do not fit in the whole figure scheme. Kindly align and adjust it with figure presentation.
Response 17: We have made changes to all the Figures in the manuscript.
Comment 18: It was strange that 1nM concentration was active inhibitor f oxidative stress while higher concentrations were not. This need to be discussed.
Response 18: The phenomenon alluded to by the Reviewer was discussed in detail in the Discussion section of the manuscript (lines 199-212).
Comment 19: What the author expects from this study in the context of future endurance, it should be mentioned in conclusion section.
Response 19: We have discussed the potential therapeutic implications of our findings in the Discussion section (lines 256-263).
